# A climate network perspective of the intertropical convergence zone

Frederik Wolf[1,2], Aiko Voigt[3,4], and Reik V. Donner[1,5]

[1]Research Domain IV - Complexity Science, Potsdam Institute for Climate Impact Research (PIK) – Member of the Leibniz Association, Potsdam, Germany
[2]Department of Physics, Humboldt University, Berlin, Germany
[3]Institute of Meteorology and Climate Research, Department Troposphere Research, Karlsruhe Institute of Technology, Karlsruhe, Germany
[4]Lamont-Doherty Earth Observatory, Columbia University in the City of New York, NY, USA
[5]Department of Water, Environment, Construction and Safety, Magdeburg–Stendal University of Applied Sciences, Magdeburg, Germany

**Correspondence:** Frederik Wolf (frederik.wolf@pik-potsdam.de)

**Abstract.** The intertropical convergence zone (ITCZ) is an important component of the tropical rain belt. Climate models continue to struggle to adequately represent the ITCZ and differ substantially in its simulated response to climate change. Here we employ complex network approaches, which extract spatio-temporal variability patterns from climate data, to better understand differences in the dynamics of the ITCZ in state-of-the-art global circulation models (GCMs). For this purpose, we study simulations with 14 GCMs in an idealized slab-ocean aquaplanet setup from TRACMIP – the Tropical Rain belts with an Annual cycle and a Continent Model Intercomparison Project. We construct network representations based on the spatial correlation pattern of monthly surface temperature anomalies and study the zonal mean patterns of different topological and spatial network characteristics. Specifically, we cluster the GCMs by means of their zonal network measure distribution utilizing hierarchical clustering. We find that in the control simulation, the zonal network measure distribution is able to pick up model differences in the tropical SST contrast, the ITCZ position and the strength of the Southern Hemisphere Hadley cell. Although we do not find evidence for consistent modifications in the network structure tracing the response of the ITCZ to global warming in the considered model ensemble, our analysis demonstrates that coherent variations of the global SST field are linked with ITCZ dynamics. This suggests that climate networks can provide a new perspective on ITCZ dynamics and model differences therein.

## 1 Introduction

One-third of Earth's precipitation falls within the narrow band of the deep tropics within 10° N/S (Kang et al., 2018). This narrow band is home to the intertropical convergence zone (ITCZ), in which the northerly and southerly trade winds of the Hadley circulation meet and give rise to surface convergence of moist air, ascent and hence rainfall (Wallace and Hobbs, 2006). Because tropical rainfall is critical to many societies and ecosystems, reliable projections of the ITCZ response to climate change are key to mitigation and adaptation efforts in a warming world (Donohoe and Voigt, 2017). Yet, global climate models are affected by persistent biases in the simulation of tropical rainfall and the ITCZ in the present-day climate, have

difficulties in capturing past ITCZ shifts, and show limited consensus on how the ITCZ location, width, and strength will change in response to increasing atmospheric carbon dioxide levels (Bony et al., 2015; Harrison et al., 2015; Byrne et al., 2018). These model difficulties reflect the importance of small-scale cloud processes and their coupling with the large-scale circulation, e.g., via cloud-radiative effects (Voigt et al., 2014) and convective mixing (Moebis and Stevens, 2012), and have motivated a large amount of theoretical and idealized work (Kang et al., 2009; Donohoe et al., 2013; Schneider et al., 2014; Biasutti et al., 2018). This has led to important insights into how the position of the ITCZ is controlled by atmospheric energy transport and sea-surface temperatures (SST).

Generally speaking, heating one hemisphere relative to the other leads to an ITCZ shift into the heated hemisphere because of the cross-equatorial atmospheric energy transport required to balance the hemispheric heating (Kang et al., 2009; Donohoe et al., 2013). Moreover, warming the surface of one hemisphere also leads to an ITCZ shift into the warmer hemisphere, in line with changes in near-surface moist static energy and boundary-layer convergence (Lindzen and Nigam, 1987; Emanuel et al., 1994). These considerations have formed our understanding of how the ITCZ is connected to the spatial pattern of atmospheric energetics and SST. These perspectives have been employed to study the movement of the ITCZ on seasonal time scales (e.g., Adam et al., 2016) as well as on the longer timescales of past and future climate change (e.g., Donohoe et al., 2013). They have further been used to understand the response of the ITCZ to specific perturbations from, for example, aerosols, ocean heat transport or clouds (e.g., Hwang et al., 2013; Frierson et al., 2013; Voigt et al., 2014, 2017) and for investigating model shortcomings in simulating the ITCZ (Adam et al., 2018). Still, in many cases the success of these perspectives is limited, and their link to the ITCZ in model simulations can be weaker than expected (Biasutti and Voigt, 2020). In the context of the present study it is also important to remember that they operate in a time-average sense: they link the time-average ITCZ position to the time-average atmospheric energy transport and time-average SST pattern. The time average can mean both a seasonal mean or a long-term annual mean.

In this study, we aim to test an alternative perspective based not on the time-average SST field but on its variability. We apply tools from complex network theory and account for the information encoded in the spatio-temporal variability of the SST pattern, which we attempt to relate to the time-average ITCZ position. Different from the frameworks described above, our network approach links spatial correlation patterns of the global SST field to the time-mean ITCZ position. For this purpose, we employ the concept of functional climate network analysis (Tsonis and Roebber, 2004; Donner et al., 2017; Dijkstra et al., 2019) that focuses on the strongest mutual statistical interdependencies among spatially distributed records of climate variability. While being based on the same correlation matrix as popular linear analysis approaches in statistical climatology such as empirical orthogonal function (EOF) analysis, this approach involves a nonlinear filter that highlights only key structures and makes their associated spatial interdependence patterns fully transparent (Donges et al., 2015b; Donner et al., 2017). As a result, functional climate networks have found a rising variety of applications in climate science (Dijkstra et al., 2019). Among others, they have provided key insights into the climate dynamics associated with large-scale tropical SST anomalies representing the oceanic manifestation of the El Niño Southern Oscillation. Specifically, the finding that climate network patterns based on global surface air temperature anomalies are significantly affected by El Niño and La Niña episodes (Yamasaki et al., 2008; Gozolchiani et al., 2008, 2011) has led to the development of improved strategies for El Niño forecasting (Ludescher et al.,

2013, 2014; Feng et al., 2016; Meng et al., 2020) and a self-consistent classification of different flavors of El Niño and La Niña based on their corresponding global imprints (Radebach et al., 2013; Wiedermann et al., 2016). Other successful applications include the development of early warning indicators for a possible collapse of the Atlantic Meridional Overturning Circulation based on ocean temperature correlations (van der Mheen et al., 2013), uncovering of key spatiotemporal patterns associated with heavy precipitation formation in different monsoon regions (Malik et al., 2012; Boers et al., 2013, 2014; Stolbova et al., 2014), improved forecasting of the Indian summer monsoon onset and rainfall amount (Stolbova et al., 2016; Fan et al., 2020), and identification of teleconnection pathways (Zhou et al., 2015; Boers et al., 2019). In all these examples, spatio-temporal climate data have been transformed into a network representation based on different concepts of statistical association used for identifying mutually dependent climate time series.

To better understand the effect of spatio-temporal SST patterns on ITCZ dynamics, we present the first application of functional climate network analysis to idealized aquaplanet simulations from the TRACMIP model ensemble (Voigt et al., 2016) that is freely available via the Earth System Grid Federation as well as the Pangeo project. The simulation setup and the corresponding data are briefly introduced in Sect. 2, which also details our analysis methodology that combines functional climate network analysis with a hierarchical clustering method to classify the TRACMIP models according to their climate network topology. We focus on two questions. First, to what extent is the network-based classification related to the models' ITCZ position in the control simulation? And second, to what extent is the response of the ITCZ to quadrupled atmospheric carbon dioxide related to changes in the climate network? The obtained results are presented in Sect. 3. The paper closes with a discussion and conclusion in Sect. 4.

## 2 Data and methods

### 2.1 TRACMIP model ensemble

The Tropical Rain belts with an Annual cycle and a Continent Model Intercomparison Project (TRACMIP) provides a suite of simulations with 14 global circulation models in an idealized aquaplanet setup and a setup with an idealized continent. TRACMIP was designed to study fundamental aspects of the ITCZ and its response to climate change, e.g., the link of the ITCZ with SST and cross-equatorial atmospheric energy transport (Biasutti and Voigt, 2020). TRACMIP can also be used in a much broader sense, including studies of phenomena such as the Arctic amplification (Russotto and Biasutti, 2020). The TRACMIP protocol has been described in detail in Voigt et al. (2016), which includes references for the participating models.

The most salient features of TRACMIP compared to other aquaplanet studies is the use of a slab ocean with a present-day-like ocean heat transport and seasonally-varying insolation. In TRACMIP, sea surface temperatures are thus interactive and the surface energy balance is closed, the ITCZ migrates north and south during the year, and the ITCZ is located in the Northern Hemisphere in the zonal-mean and time-average, consistent with the present-day climate.

In this work, we use aquaplanet simulations performed by all 14 models contributing to the intercomparison project. The AquaControl simulation is run with a present-day like $CO_2$ concentration of 348 ppmv. In the Aqua4xCO2 simulation, $CO_2$ is quadrupled to 1392 ppmv, leading to a model-dependent increase of global-mean surface temperature by 3-10 K, and changes in

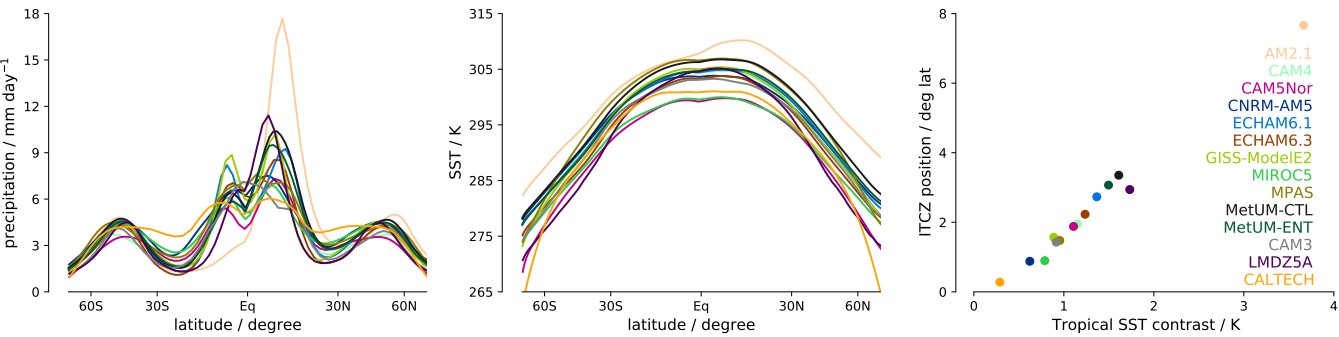

**Figure 1.** Time-average zonal-mean precipitation (left) and surface temperature (middle) in the AquaControl simulations. The right plot shows the tight correlation of the time-average ITCZ with the SST contrast between the Northern and Southern Hemisphere tropics.

the ITCZ location that range from a slight southward shift to strong northward shifts by up to $8°$ in latitude. Following Biasutti and Voigt (2020), the ITCZ position is calculated by the precipitation centroid as defined in Adam et al. (2016) over latitudes within $20°$ N/S. This locates the ITCZ somewhat closer to the equator compared to the values diagnosed in Voigt et al. (2016), but the results of our analysis are insensitive to the definition of the ITCZ position. Figure 1 illustrates the precipitation and sea surface temperature pattern of the Aquacontrol simulations. This figure also demonstrates the tight correlation of the ITCZ position with the tropical SST contrast (correlation coefficient $> 0.99$). Again following Biasutti and Voigt (2020), the latter is defined as the SST difference between the Northern and Southern Hemisphere tropical means (Eq.–$25°$ N and $25°$ S–Eq., respectively).

Figure 1 confirms that SST is a prime control on the ITCZ, a fact that has been exploited in many previous studies. However, in contrast to previous work that used the time-average SST, we here apply functional climate network representations to study the relation between the ITCZ and the internal variability of the SST field. An illustration of the temporal SST variability is shown in Fig. 2 for the AquaControl simulation. The SST variability is minimal near the equator, is relatively constant in the subtropics and midlatitudes, and drops off near the poles. This pattern is broadly in line with the variability in the sum of surface turbulent fluxes (sensible and latent heat flux) and surface winds (not shown).

The TRACMIP model output is provided on regular latitude-longitude grids that have a model-dependent horizontal resolution ranging from 1 to $3°$ in latitude and longitude. For the AquaControl simulations, we restrict our analysis to the last 30 years and for the Aqua4xCO2 simulations to the last 25 years to ensure that the models are in statistical equilibrium. As described below, we focus on monthly-mean SST fields, from which we construct functional climate networks and study their relation to the time-average ITCZ position and tropical SST contrast.

## 2.2 Functional climate networks

Functional climate networks constitute an application of complex network theory to understand functional relations in the Earth's climate system. In general, complex networks often serve as abstract mathematical models of complex systems, in which individual entities are represented by nodes that are connected by links symbolizing interdependencies among the enti-

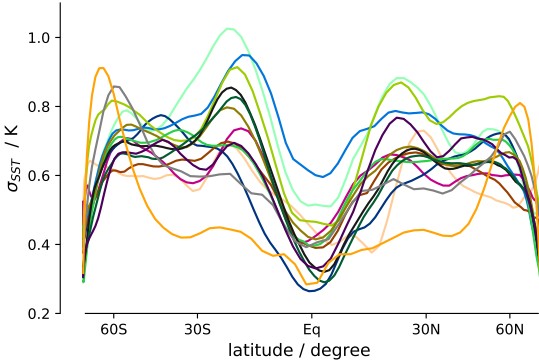

**Figure 2.** Zonal-mean of the temporal standard deviation of monthly-mean SST in the AquaControl simulations. The models are color coded as in Fig. 1.

ties. In climate networks nodes are identified with geographical locations at which climate variability information is available in the form of time series (in our case, the grid points of climate model outputs), and links connect pairs of nodes whose

climate time series exhibit a certain level of statistical association. In this work, we analyze the set of SST time series on the longitude-latitude grid separately for each model and simulation run using the methodological setup described in the following.

First, we calculate the linear Pearson correlation coefficient for the monthly SST anomalies of all pairs of grid points. The time series of the SST anomalies are computed by subtracting the monthly-mean climatology from the original SST time series. In principle, we could utilize any statistical association measure in this step. We chose the Pearson correlation here as it has

been successfully applied in previous studies (Donges et al., 2009; Radebach et al., 2013; Wiedermann et al., 2016) and is suitable for relatively short time series (here, 300 to 360 monthly anomaly values). For a model output with $N$ grid points, this leads to a correlation matrix $\mathbf{S}$ of dimension $N \times N$.

Second, we threshold the correlation matrix and transform it into a binary matrix $\mathbf{A}$. $\mathbf{A}$ is the adjacency matrix of the associated climate network representation of the underlying data set and has the same dimension as $\mathbf{S}$. If the correlation

value $s_{ij}$ between two grid point time series (nodes) $i$ and $j$ is larger than the prescribed threshold, we set the corresponding matrix element of $\mathbf{A}$ to $a_{ij} = 1$ (i.e., there is a link between nodes $i$ and $j$), and to $a_{ij} = 0$ otherwise (no link). We choose the correlation threshold in such a way as to obtain a link density of $\rho = 0.005$. This implies that the network only includes the strongest $0.5\%$ of all possible $N(N-1)/2$ undirected links (when excluding self-correlations, i.e., $a_{ii} = 0$). For global networks with a comparable resolution, this order of magnitude has been a common choice in previous studies (Radebach

et al., 2013; Donner et al., 2017). It has to be noted, however, that changes in the link density of a climate network have been previously shown to potentially result in qualitatively different behaviors of the resulting network characteristics (Radebach et al., 2013; Wiedermann et al., 2017b) (depending on the specific network property under study). For this reason, we keep the link density fixed for all analyses performed in this work. However, since the individual models differ in their spatial grid resolution, the total number of links will differ between models, with a lower number of links in case of coarser grids. This has

direct consequences for the quantitative analysis of the resulting networks. In the following, we will choose our analysis setup such that the effect of a different number of links on the final results is minimized.

Third, based on the adjacency matrix we calculate the values of two network characteristics: the degree and the average link distance. The degree $k_i$ encodes the number of links of each node $i$,

$$k_i = \sum_{i=1}^{N} a_{ij}, \tag{1}$$

where the index $i$ corresponds to a specific latitude-longitude grid point. As our climate networks are spatially embedded on regular spherical grids, the spatial density of nodes on a spherical surface is not homogeneous and increases towards the poles. To make all nodes equally representative, we utilize the concept of the so-called node splitting invariance (n.s.i., Heitzig et al. (2012)) and define the accordingly node-weighted n.s.i.-degree as

$$k_i = \sum_{i=1}^{N} w_i a_{ij}. \tag{2}$$

Here, $w_i$ is the n.s.i. node weight, which in our case is given by the cosine of the latitude. For simplicity, in the remainder of this work, we will briefly refer to the n.s.i. degree (sometimes also termed area-weighted connectivity (Tsonis et al., 2006; Tsonis and Swanson, 2008)) as the degree. Moreover, instead of discussing the full spatial pattern of degree on the whole sphere, we will focus on the zonal-mean degree. The meaningful calculation and interpretation of this property are ensured by the zonally-uniform boundary conditions of the aquaplanet setup. To compare the zonal-mean degree pattern between models despite their differences in grid resolution, we normalize the zonal-mean degree such that its sum over all latitudes is 1.

The information provided by the purely topological measure of node degree is complemented by the average link distance, which is a spatial (geometric) characteristic. The average link distance is the mean great circle distance between a given node and all its connected nodes (i.e., the mean spatial length of all links of a given node). For simplicity, the average link distance is given here in units of radians; physical distances could be easily obtained by scaling the values with the Earth's radius.

We emphasize that we have also studied a suite of additional local (such as the local clustering coefficient) as well as global network properties (such as the network transitivity) (Radebach et al., 2013; Donner et al., 2017). These measures have not provided additional insights and are therefore not reported in the following.

## 2.3 Hierarchical Clustering

After having constructed the network for each individual model and simulation, we study to what extent the models can be grouped according to their resulting zonal-mean network measure patterns. For this purpose, we employ a hierarchical cluster analysis.

As a first step, we resample the zonal-mean network measure distribution to the lowest latitudinal resolution of all models (CALTECH, $2.8°$) utilizing linear interpolation. Then, we calculate the Pearson correlation coefficient between (selected parts of) the values of the resampled zonal-mean network measures as a function of the latitude between all pairs of models. Since the Pearson correlation is invariant under rescaling and possible additive terms, offsets between the individual models' magnitudes of the zonal mean network measures that originate from the different grid resolutions do not impact the correlation values.

We calculate the Pearson correlation individually for both the degree and the link distance and then sum the correlation values for each model pair. This results in a matrix $C$ of dimensions $14 \times 14$ with elements representing the pairwise similarity between all 14 models in terms of both types of zonal-mean network measures. The similarity measure can therefore exhibit values in the range $[-2, 2]$ (i.e., the sum of two values each bounded by $[-1, 1]$). For convenience, we transform this into normalized values within the interval $[0, 1]$ by employing a linear rescaling

$$c_{ij}^{new} = \frac{c_{ij} - \min C}{\max\limits_{ij}(c_{ij} - \min C)},$$

with $C_{new}$ being the rescaled inter-model similarity matrix that is then used as an input for the hierarchical cluster analysis. Note that $\min C = \min_{ij} c_{ij}$ and $\max C = \max_{ij} c_{ij}$ are the minimal and maximal values among all inter-model correlations and that $\min C$ can hence be negative.

Finally, we cluster models by means of the hierarchical cluster analysis as implemented in the python package `scipy`, where we use the single linkage method for successively combining groups of models according to the highest pairwise correlation among the respectively included models. This methodological choice ensures that the most similar models are grouped into the same cluster, yet allows for the possibility to produce outliers. With the number of considered models being quite low, we can at every iteration of the algorithm identify such outliers and explain this by the properties of the employed cluster analysis method while ensuring that the most similar models indeed end up in the same cluster. Specifically, based on the models' mutual similarity, our hierarchical cluster analysis method iteratively identifies pairs of models that are successively merged into clusters until all models belong to a single cluster. The order of this clustering is depicted in a dendrogram. In the visual representation used in this work, the resulting dendrograms are meant to be read from left (where each individual model constitutes a single cluster) to right (all models are combined in one cluster). Vertical lines indicate a merge of two models or clusters of models, while the horizontal lines represent the increasing cophenetic distance between the clusters (capturing the degree of similarity between the two groups based on the rescaled inter-model similarity matrix). Cutting the dendrogram at a certain level of similarity, i.e., cophenetic distance, leads to certain clusters of models that will then be used in the remainder of this study.

We are aware that there are alternative ways both to quantify the similarity between different models based on the values of the different network measures and to cluster the models. Our methodological choices reflect the need to account for differences in the grid resolution of the 14 global climate models, and our preference for a simple and intuitive analysis setup.

## 2.4 Robustness tests

We acknowledge that our analysis setup is based on several specific choices of methodological parameters or variants. To address this, we have tested our results for robustness in the following manner. Due to the limited amount of data and only 14 models to be clustered, we have performed two basic tests on the results. First, we have split the 30 years into two 15-year periods and have analyzed the two sets of anomaly time series individually. Although the results are neither identical nor completely match the results from the analysis of 30 years, the clustering of the models is only marginally affected (see Supplementary Material Section S1). Second, we have stacked the time series of 20 randomly chosen years and compared

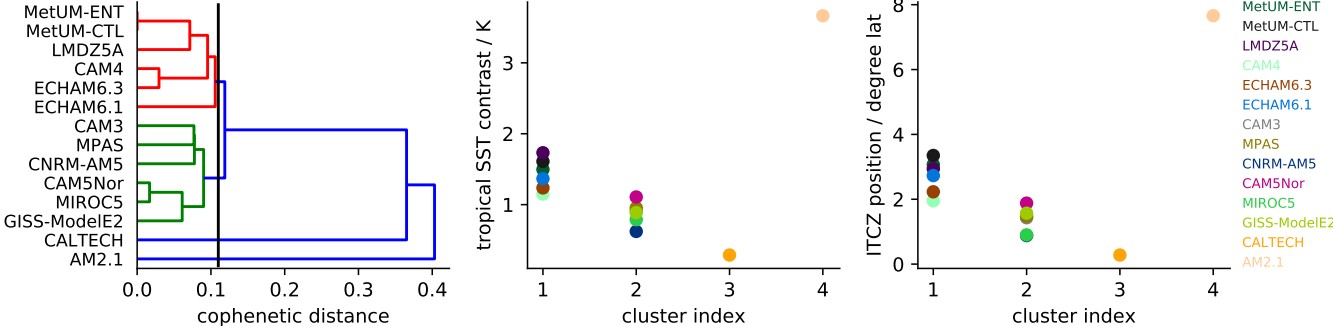

**Figure 3.** Model clustering of the AquaControl global networks (left) along with the tropical SST contrast (middle) and ITCZ position (right) for the four identified clusters. The left panel shows the dendrogram obtained from the clustering of the zonal-mean network measures. The vertical line indicates the level of cophenetic distance at which we split the models into four clusters.

the resulting zonal-mean network measures for 20 independent realizations. We have found that the main features of the zonal-mean degree distribution are retained on average among the resulting network ensembles (see Supplementary Material Section S2). The two sensitivity tests hence indicate that the results described below are robust.

Regarding the cluster analysis employed for grouping the TRACMIP models according to the mutual similarity of their respective network representations, there exists a large body of alternative methodological approaches, including hierarchical clustering techniques with different linkage strategies (e.g., complete linkage, average linkage, or Ward's method), partitioning methods like $k$-means, spectral clustering, and many more. To account for the corresponding variety of options and highlight the relevance of an educated choice of the method, we have also considered the complete and average linkage methods (see Supplementary Material Section S3). The resulting commonalities and differences with respect to single linkage clustering will be discussed for the example of the AquaControl simulations.

## 3  Results

In the following, we present the results of our functional network analysis. We first make use of the AquaControl simulations and start by investigating climate networks constructed from the complete (global) SST field that include both, tropical and extratropical nodes (Sect. 3.1). Subsequently, we study networks for which some of the regional connections were excluded (Sect. 3.2). Finally, we study the response of the climate networks and the ITCZ position to a quadrupling of $CO_2$ by repeating the analysis for the Aqua4xCO2 simulations (Sect. 3.3).

### 3.1  AquaControl simulations: Global network properties

The global network analysis is shown in Fig. 3 and separates the models into four clusters (left panel), with cluster 1 and 2 each consisting of 6 models, and cluster 3 and cluster 4 each containing only a single model. The zonal-mean network measures that underlie this clustering are shown in Fig. 4 and discussed in more detail below. Importantly, Fig. 3 demonstrates

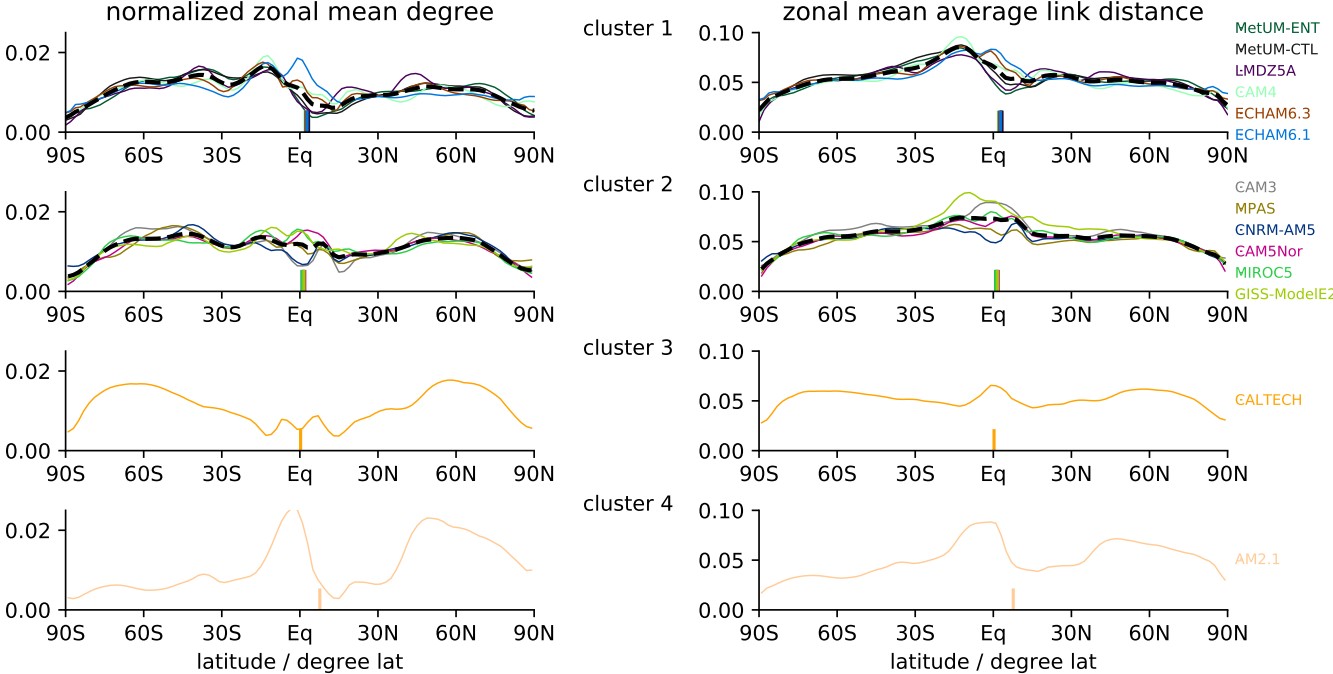

**Figure 4.** Zonal-mean network measures from the global networks of the AquaControl simulations for each of the four clusters. The left panels show the normalized zonal-mean degree, the right panels show the corresponding average link distance patterns. Vertical lines indicate the position of the ITCZ. The cluster-mean is shown by black dashed lines.

that the clustering successfully separates models in terms of their time-average tropical SST contrasts (central panel) and, to a slightly lesser extent, ITCZ positions (right panel). Specifically, the models in cluster 1 exhibit a stronger SST contrast and a more poleward shifted ITCZ than the models in cluster 2. Indeed, there is no overlap between the two clusters regarding the respective SST contrast and ITCZ position. The clustering also separates models in terms of the strength of their Southern Hemisphere Hadley circulation as measured by the magnitude of the minimum of the mass stream function (cluster 1: 130-158; cluster 2: 102-129; cluster 3: 53; cluster 4: 266; all values in units of $10^9$ kg/s). This is expected since the ITCZ position and the Hadley cell strength tend to be strongly correlated (Donohoe et al., 2013).

The zonal-mean network measures of the four clusters are presented in Fig. 4, where the left panels show the normalized degree and the right panels the average link distance. The spatial patterns of the zonal-mean network measures differ systematically between the clusters along with their respective ITCZ positions and SST contrasts. This consistency demonstrates that the hierarchical clustering is indeed climatologically meaningful. Despite some inter-model variability in each cluster, clusters 1 and 2 exhibit systematic differences. For cluster 1 (Fig. 4, first row), all models show a coherent degree and link distance minimum around the position of the ITCZ and a marked peak of both measures related to a strong Southern Hemisphere Hadley cell. This finding has already been reported by Wolf et al. (2019) and reflects the fact that the models of cluster 1 have the strongest Southern Hemisphere Hadley cell across the model ensemble. For cluster 2, the zonal-mean average link

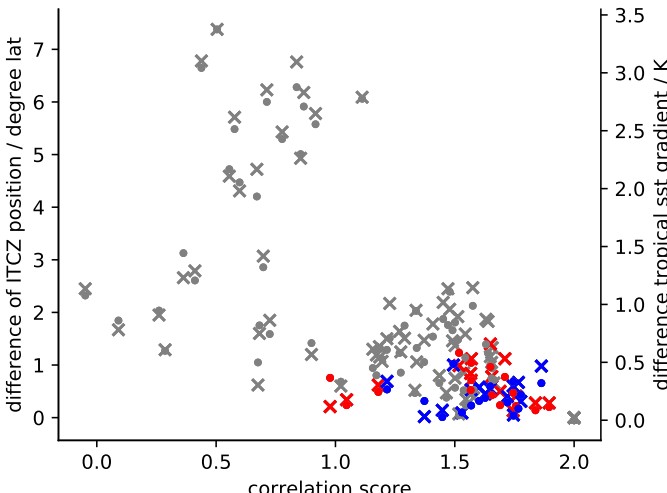

**Figure 5.** Pairwise inter-model correlation score versus the difference of ITCZ position and SST contrast of all model pairs. Model pairs where both models fall into the same multi-model cluster are colored in red (cluster 1) and blue (cluster2), respectively. The difference of the ITCZ position is marked by circles, the difference in the SST contrast is marked by "x".

distance exhibits a broad maximum around the ITCZ position instead of the minimum found for cluster 1 (Fig. 4, second row). Moreover, the zonal-mean network measures of cluster 2 are more symmetric with respect to the equator than for the models in cluster 1. The comparison between clusters 1 and 2 reveals that a more northward ITCZ position tends to be associated with less symmetric network properties and minima in degree and link distance near the ITCZ, whereas an ITCZ closer to the equator is accompanied by a more symmetric pattern of zonal-mean network characteristics with only small meridional

contrasts in the tropical degree and a near-equatorial maximum in the average link distance.

As opposed to the aforementioned two groups of models, the networks derived from the CALTECH and AM2.1 models, which form cluster 3 and cluster 4, respectively, show zonal-mean network measures that resemble extreme versions of cluster 1 (for AM2.1) and cluster 2 (for CALTECH). CALTECH (cluster 3) has the ITCZ closest to the equator among all models, and its network characteristics exhibit near-perfect symmetry with respect to the equator. By contrast, for AM2.1 (cluster 4) the

ITCZ is shifted far into the Northern Hemisphere and its zonal-mean network characteristics exhibit hardly any symmetry with respect to the equator, yet marked minima of degree and average link distance at about the latitudinal position of the ITCZ.

We further demonstrate the success of the functional climate network analysis along with the hierarchical cluster analysis in the following manner. For all pairs of models, we plot the inter-model difference in the ITCZ position and tropical SST contrast as a function of the similarity of the models' zonal-mean network characteristics. The latter is measured by the elements of

the original (non-rescaled) inter-model similarity matrix $c_{ij}$, which represent the combined similarity of the patterns of the two considered zonal-mean network characteristics (see Sect. 2.3). Here, we note again that $c_{ij}$ is bounded to the interval $[-2, 2]$ because the values of the Pearson correlation coefficients for zonal-mean degree and zonal-mean average link distance are both

restricted to $[-1, 1]$. Figure 5 shows the resulting scatter plots of the pairwise similarity coefficient versus the ITCZ position and tropical SST contrast, respectively.

Despite the generally large scatter, there is a clear tendency towards smaller differences in the ITCZ position and SST contrast for models whose climate networks are more similar (Fig. 5). Put differently, this underlines that for models that are identified as more similar by the network and cluster analysis (larger correlation score on the x-axis), the pairwise difference in the ITCZ position and SST contrast tends to be smaller than for models with less similar network characteristics. In addition, all combinations of model pairs that belong to cluster 1 or cluster 2 (recall that cluster 3 and cluster 4 only include only one model each) are colored in red and blue, respectively. This shows that indeed the clustering only groups models together that have similar network characteristics.

In the analysis described above, we have utilized a hierarchical cluster analysis with the single linkage approach. In general, there exist a plethora of other methodological options for clustering entities. Among the hierarchical approaches, the existing methods differ in their respective criteria for merging elements or groups of elements. For example, one common alternative to the single linkage approach is the complete linkage method, where the largest (in our case cophenetic) distance among pairs of elements is considered. By contrast, the average linkage technique evaluates the mean of all pairwise distances among models belonging to two different groups. An application of both mentioned alternative linkage methods leads to substantial differences in the resulting group structure, although certain pairs of models are categorized into the same subgroup as under the single linkage method (see Supplementary Material Section S3). This finding could be a direct consequence of the specific distribution of similarity scores (see Fig. 5). The minor differences in mutual similarity among the models in the single linkage-based clusters illustrate the sensitivity of a clustering procedure considering the mean of all pairwise similarities (where single model pairs with reduced similarity might essentially influence the average group similarity and the largest cophenetic distance between group members). As already mentioned, there would be many further alternatives of hierarchical or non-hierarchical clustering methods beyond the two aforementioned ones. In general, we can expect that due to the rather small size of the studied model ensemble, it is of minor interest for the purpose of this study to analyze multiple clustering criteria more systematically (seeking for some abstract statistical optimality). We have therefore chosen to focus on the hierarchical single linkage approach in all following analyses, since it allows for a straightforward interpretation of the resulting dendrograms.

In summary, our above results show that functional climate network analysis, although only using information from monthly variability of the global SST field, is able to distinguish time-average model differences in the ITCZ position, SST contrast, and Hadley cell strength.

## 3.2 AquaControl simulations: Networks with stepwise exclusion of regional connections

The global network analysis in the previous subsection included both tropical and extratropical connections. To disentangle the relative importance of tropical and extratropical connections, we repeat the analysis but successively exclude different classes of links by setting the corresponding elements of the adjacency matrix to zero. Notably, this strategy removes *links* from the network whereas the zonal network measures are attributes of *nodes* and each link contributes to the degree and average link distance of two different nodes. Hence, we still retain complete zonal-mean characteristics of networks with a specific subset of

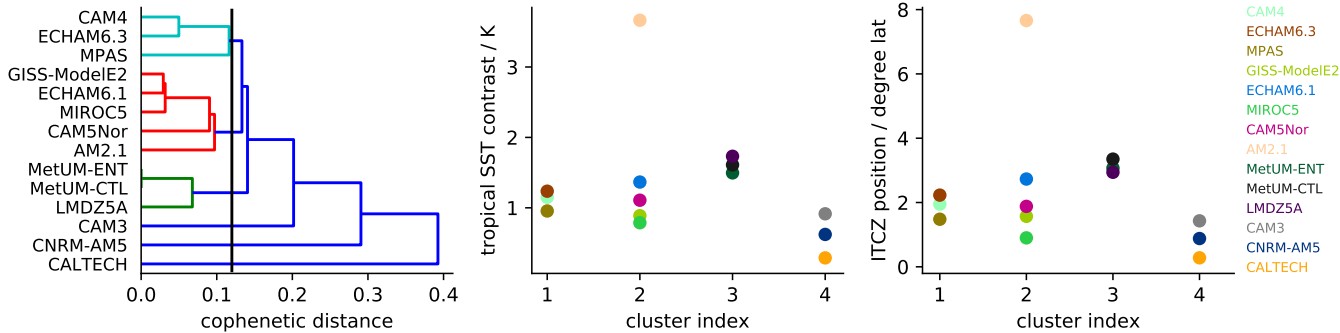

**Figure 6.** Model clustering based on the AquaControl networks without extratropical–extratropical connections (left) and values of the tropical SST contrast (middle) and ITCZ position (right) for all models of the four identified clusters. The left panel shows the dendrogram obtained from the hierarchical clustering of the zonal-mean network measures. The vertical line indicates the level of cophenetic distance at which we split the models into four clusters.

links. The analysis of the residual network structures thereby allows us to quantify the importance of, e.g., connections within the tropics or between the tropics and the extratropics.

First, we remove all trans-equatorial connections between the Northern and Southern Hemisphere extratropics ($> 35°$ N/S).
This analysis leads to almost indistinguishable results compared to the global networks (not shown), because the number of trans-equatorial extratropical-extratropical connections is very low for all models. This shows that inter-hemispheric teleconnections between the Northern and Southern Hemisphere extratropics do not markedly affect the ITCZ position.

Second, we exclude all extratropical-extratropical connections (i.e., also links connecting two nodes within the extratropics of the same hemisphere). As a consequence, the resulting network characteristics in the extratropics only include tropical-
extratropical connections, while the network measures in the tropics feature both tropical-tropical and tropical-extratropical connections. This leads to a very low link density in the extratropics while the link density in the tropics remains almost as large as in the analysis of the complete network. The extratropics of both hemispheres together account for $110°$ in latitude that contribute to the zonal mean values (with low link density and an irregular distribution), while the tropics cover only $70°$ in latitude (but with high link density and smooth distribution). As a consequence, correlations between the irregular distributions
in the extratropics do not lead to meaningful results, while the patterns of zonal-mean network measures in the tropics also include the information from extratropical-tropical links. In the following, we therefore only consider zonal-mean network characteristics between $35°$ S and $35°$ N to quantify the similarity between the respective model outputs. By considering the zonal-mean characteristics only for this tropical band, we obtain smooth and stable patterns that represent both inner-tropical and tropical-extratropical connections.

The dendrogram obtained by our hierarchical cluster analysis as well as the associated SST contrasts and ITCZ positions are shown in Fig. 6. Unlike for the complete network, the analysis without extratropical–extratropical connections separates the models into 4 clusters of similar size. 3 of the 4 clusters differ regarding their SST contrast, ITCZ position, and Southern Hemisphere Hadley cell strength. The network analysis thus identifies some model differences in the tropical climate also

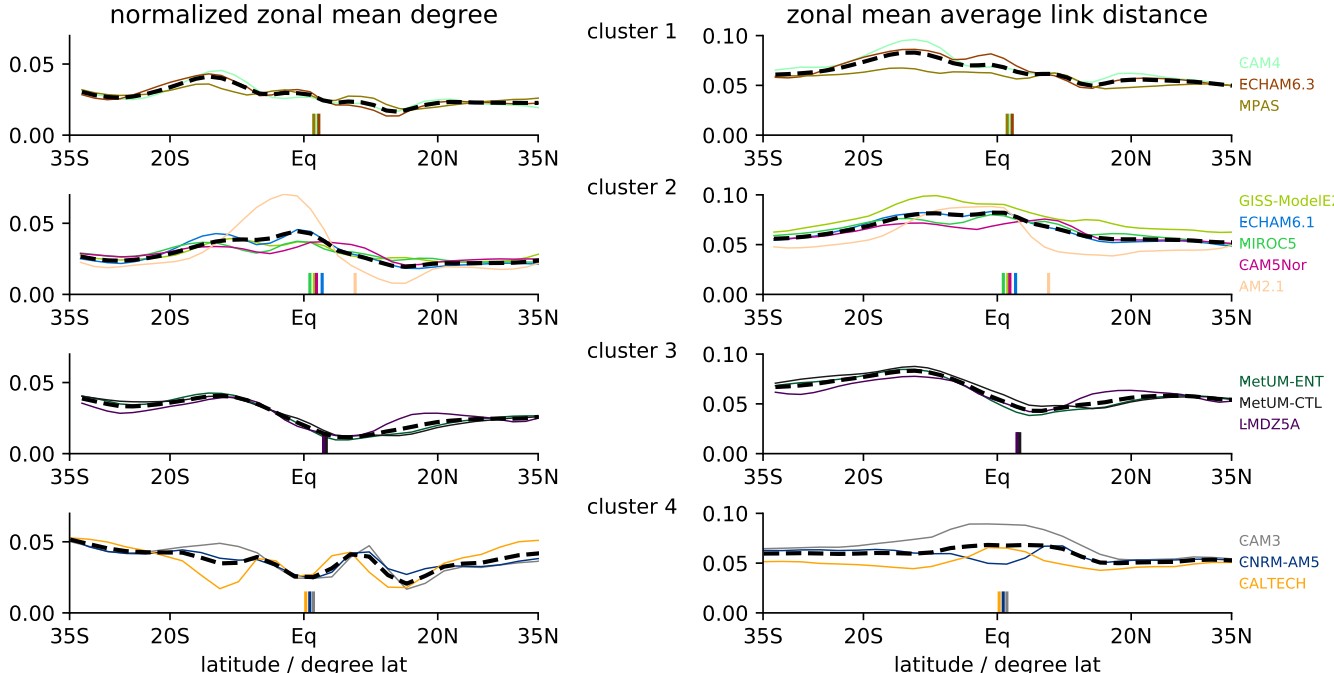

**Figure 7.** Zonal-mean network characteristics for the AquaControl simulations without extratropical-extratropical connections for each of the four clusters of models. The left panels show the normalized zonal-mean degree, while the right panels display the corresponding average link distances. Vertical lines indicate the respective position of the ITCZ. The cluster means are shown by black dashed lines.

when extratropical–extratropical connections are removed, but the success in doing so is reduced compared to the analysis of

the global network.

The zonal-mean network measures underlying the clustering of Fig. 6 are shown in Fig. 7. The models in cluster 3 (third row) exhibit minima of the zonal-mean network characteristics near the ITCZ and maxima in the region of the Southern Hemisphere Hadley cell. A similar feature is to some extent visible for the models in cluster 1 (first row), although it is blurred by additional features like additional local minima and marked maxima of both network properties in the region of the Northern Hemisphere

Hadley cell. The network measures for models in cluster 2 (second row) exhibit rather heterogeneous distributions, consistent with the relatively large model spread in SST contrast and ITCZ position in that cluster. Finally, the network properties of the models in cluster 4 (bottom row) feature some level of symmetry with respect to the equator, although their distributions differ substantially and their cophenetic distance in the dendrogram is large (Fig. 6 left). In line with the previous results, we notice that a decrease in the level of symmetry in the network measures is associated with an ITCZ that is more strongly shifted into

the Northern Hemisphere. This is expected as for the ITCZ to be shifted away from the equator, there needs to be some level of hemispheric asymmetry in the SST pattern.

Finally, we also tested the effect of additionally excluding tropical–extratropical connections, thereby retaining only inner-tropical connections. In this case, we did not find coherent clusters with distinct ITCZ positions (not shown). This indicates

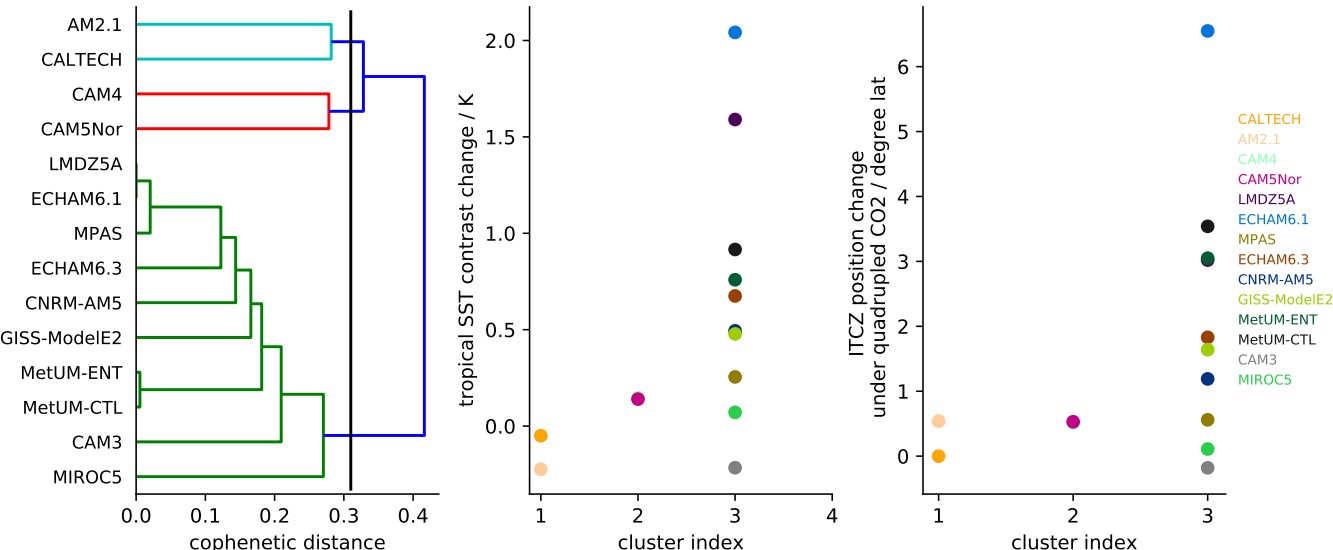

**Figure 8.** Model clustering based on the difference between the Aqua4xCO2 and AquaControl global networks (left) along with the warming induced changes in the tropical SST contrast (middle) and ITCZ position (right) for the three identified cluster. The left panel shows the dendrogram obtained from the clustering of the difference in the zonal-mean network characteristics. The vertical line indicates the level of cophenetic distance at which we split the models into the three clusters.

that tropical–extratropical interactions are important for the ITCZ dynamics (Kang, 2020), but different from time-mean frame-
works which typically find a good relation between tropical SST and the ITCZ position (e.g., Donohoe et al., 2013; Biasutti et al., 2018). Furthermore, we have tested our method based on zonal-mean SST fields. Again, this has not led to an inter-pretable clustering, and the strong decrease in the number of grid points due to the zonal averaging actually turned out to be a challenge to our analysis method. One possible explanation for this is that zonal asymmetries in the tropical circulation, e.g., those related to tropical waves, impact the ITCZ position (Biasutti et al., 2018) and are captured by the network approach,
provided that the latter is fed with latitude-longitude data.

### 3.3   Aqua4xCO2 - AquaControl simulations: Climate response to quadrupling carbon dioxide

In the previous two subsections, we have demonstrated that functional climate network analysis of monthly SST anomalies can identify model differences in the time-average SST contrast and ITCZ position based on the AquaControl simulations. In the following, we study if the analysis can also help to understand model differences in the ITCZ response to global warming
triggered by an increase in atmospheric carbon dioxide content in the Aqua4xCO2 simulations. Across the model ensemble, the ITCZ response varies from a slight southward shift by less than $1°$ in latitude to a strong northward shift by up to $8°$.

We analyze the change in the network measures between the AquaControl and the Aqua4xCO2 simulations. We compute the difference between the zonal-mean network characteristics of AquaControl and Aqua4xCO2 for each model, from which we perform the similarity-based hierarchical cluster analysis. The corresponding results are summarized in Figs. 8 and 9. It can be

seen that our analysis identifies three clusters. These are presented together with the warming induced changes in the tropical SST contrasts and ITCZ positions in Fig. 8. Cluster 1 and 2 each consist of two models, and both show a reduced ITCZ response as compared to the other models. Cluster 3 contains the majority of models (10 out of 14) and spans the entire ensemble range of ITCZ responses. This wide spread indicates that in contrast to the control climate, the combination of functional network and hierarchical cluster analysis does not pick up model differences in the ITCZ response to global warming. This finding

is further illustrated in Fig. 9, which shows the differences in the zonal-mean network properties between the AquaControl and Aqua4xCO2 simulations. While clusters 1 and 2 again both exhibit muted ITCZ responses, they differ substantially in the specific response of their network characteristics. Likewise, for cluster 3, there is no common spatial pattern in the response of the network measures.

In general, it can be noted that the network analysis appears to fail capturing inter-model differences in the ITCZ response

to warming. In contrast to simulations with realistic present-day boundary conditions, for which the zonal variations in the tropical rainfall response to warming can make it difficult to extract a meaningful zonal-mean response, TRACMIP aquaplanet simulations employ zonally-symmetric boundary conditions. The failure of the network analysis thus must have a different origin and indicates that the climate change response of the SST networks is *not* tightly linked to the ITCZ climate change response. Within the conceptual limitations of the present analysis, the reasons for this behavior remain unclear to us. One

possibility could be that unravelling the climate change response would require a different network representation that involves other atmospheric fields in addition to SST, e.g., changes in the vertical profile and gross moist stability of the tropical atmosphere, which Biasutti et al. (2018) suggested to play an important role in TRACMIP. Corresponding follow-up investigations are outlined as a possible subject of future work.

## 4 Discussion and Conclusions

The ITCZ is a central element of Earth's climate, yet understanding its dynamics and anticipating its response to climate change remains a challenge. In this work, we have proposed a new perspective on the ITCZ by means of complex network theory. We have tested this perspective by analyzing a multi-model ensemble of idealized aquaplanet simulations provided by TRACMIP. The main difference between our work and previous considerations of the ITCZ is that our perspective is based on monthly-mean anomalies of sea-surface temperature (SST), whereas previous work focused on the relation of the time-averaged ITCZ

to time-average SST and atmospheric energy transport, where the time-average can be seasonal, yearly or longer.

We have constructed complex network representations based on the correlation pattern of monthly SST anomalies in the control simulation (AquaControl) and a global-warming simulation (Aqua4xCO2). We found that the zonal-mean node degree and average link distance of functional climate networks can separate models in terms of their ITCZ position, SST contrast and Hadley cell strength in the control simulation. This separation also holds when extratropical–extratropical connections are

370 excluded, but breaks down when further connections are excluded. This shows that the network approach correctly identifies that extratropical-tropical connections are important in setting the tropical climate (Kang, 2020). The network analysis is also consistent with known mechanisms such as a strong correlation between Hadley cell strength and ITCZ position. Overall, the

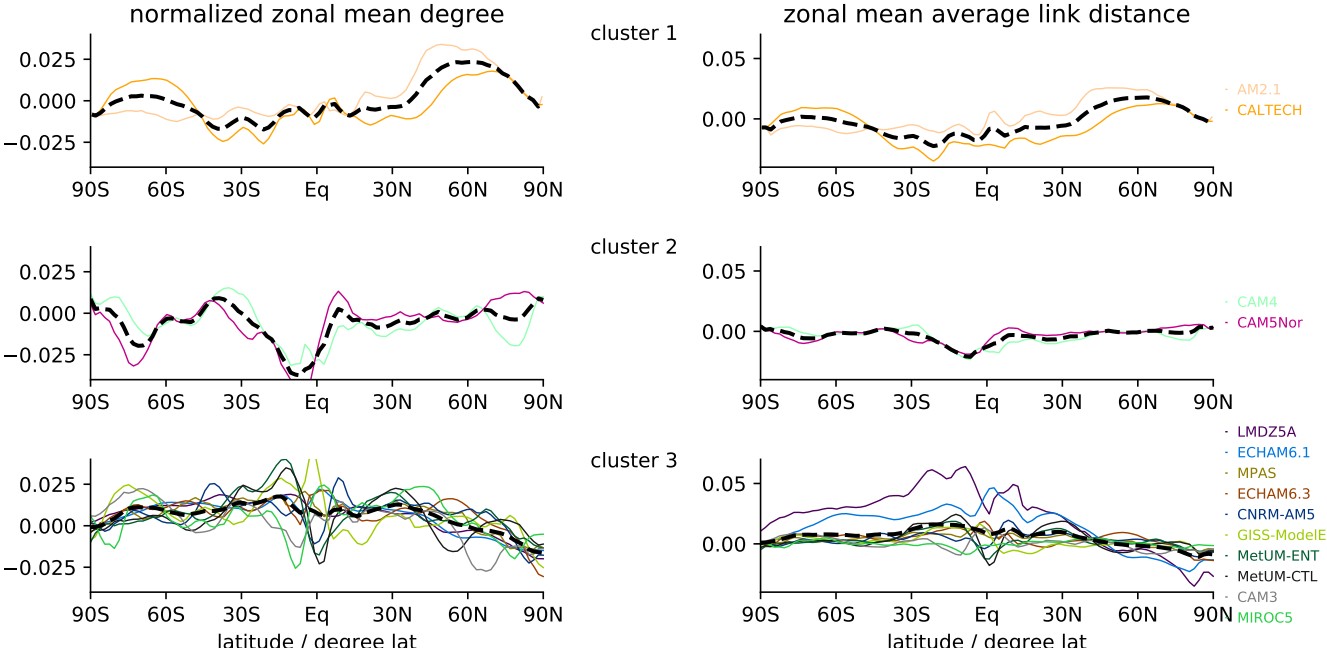

**Figure 9.** Differences in the zonal-mean network properties of the AquaControl versus Aqua4xCO2 global networks for the three identified clusters. The left panels show the normalized difference between the zonal-mean degrees of the AquaControl and Aqua4xCO2 based networks, while the same is shown in the right panels for the respective average link distances.

climate network analysis indicates that the time-mean ITCZ is connected to spatiotemporal variability of the monthly SST, a finding that is not obvious from previous work. However, we also note that the network analysis was unable to separate model differences in the ITCZ response to global warming.

Because the aquaplanet setup is zonally symmetric in a statistical sense, we have restricted our analysis to zonal mean network properties. This simplification implied the loss of local (single-node) information, which prohibited us to identify possible local connectivity structures and fully exploit the complete distribution of links in the network. For example, we have not analyzed if and how individual nodes are connected in the meridional and zonal direction. Future work could look at this aspect in more detail to reveal the role of, e.g., zonally-propagating tropical waves associated with large-scale patterns of SST, wind and atmospheric energy transport. This could also involve other network characteristics like betweenness centrality (Donges et al., 2009) or edge directionality (Wolf et al., 2019), and would complement the traditional empirical orthogonal function (EOF) analysis or more sophisticated pattern recognition approaches such as self-organizing maps (SOM).

One central goal of this study was to investigate to what extent correlation-based functional climate networks provide insights into the ITCZ dynamics as simulated in state-of-the-art global climate models. Some of the results might seem unsurprising, e.g., the fact that an ITCZ closer to the equator tends to be associated with a more hemispherically symmetric network. In some sense, however, such expected results are reassuring, as they hint at the fact that the network analysis is plausible. Although

we have discussed several of the emerging network measures in a climatological context, their implications and specific links to the dynamics of the atmosphere and climate system have remained to be further explored. This could include a refinement

of the functional network approach to take into account time leads and lags between the ITCZ and regional SST variability. For example, following extratropical perturbations, previous work showed that the extratropical SST response leads those of tropical SST and the ITCZ by several months or even longer, depending on the ocean heat capacity (Woelfle et al., 2015). The network methodology can also be extended in a straightforward manner to include other variables (Donges et al., 2011; Wiedermann et al., 2017a), such as the stratification of the tropical atmosphere and energy fluxes, and to reflect possible lead-

lag relationships between the ITCZ and the energy budget of the extratropical atmosphere (Adam et al., 2018). In summary, we therefore consider this study a first step towards applications of climate networks, and more broadly, topological data analysis to understand fundamental climate dynamics.

*Code and data availability.* The python package `pyunicorn` (Donges et al., 2015a) used for the functional climate network generation and analysis is freely available at https://github.com/pik-copan/pyunicorn. The TRACMIP data sets are available in cmorized format via the

Earth System Grid Federation that also holds CMIP data and is also available via the Pangeo project (https://pangeo.io).

*Author contributions.* All authors contributed to the design of the study. FW implemented the network analysis and analyzed the data. All authors analyzed the results and wrote the manuscript. All authors read and approved the final manuscript.

*Competing interests.* The author declare no competing interests.

*Acknowledgements.* This work has been supported by the IRTG 1740/TRP 2011/50151-0 (funded by the DFG and FAPESP). AV has re-

ceived financial support by the German Ministry of Education and Research (BMBF) and FONA: Research for Sustainable Development (www.fona.de) under Grant Agreement 01LK1509A. AV further acknowledges funding by NSF award AGS-1565522. RVD acknowledges funding by the German Federal Ministry for Education and Research (BMBF) via the BMBF Young Investigators Group CoSy-CC$^2$ (grant no. 01LN1306A), the Belmont Forum / JPI Climate project GOTHAM (grant no. 01LP16MA) and the JPI Climate / JPI Oceans project ROADMAP (grant no. 01LP2002B).

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
