# Peer review of "A climate network perspective of the intertropical convergence zone"

_Earth System Dynamics, 2020_

## Referee Comment (RC1) · Anonymous Referee #1 · 1 Dec 2020

Review of

**A climate network perspective of the intertropical convergence zone**

Frederik Wolf, Aiko Voigt, and Reik V. Donner

**General**

The authors use 'functional climate network representations' to study the relation between the ITCZ and the SST field. The uniqueness of the approach is that, unlike previous studies that considered the climatological relation between these fields, the analysis based on climate network uses the internal variability of the SST field (based on monthly means). The analysis is performed on the Tropical Rain Belts with an Annual cycle and a Continent Model Intercomparison Project (TRACMIP), which includes 14 GCMs with idealized ocean components. Being based on idealized models, TRACMIP suffers from several limitations which are noted in the text. The analysis, therefore, serves as a 'proof of concept' which can later be applied to more realistic datasets. My understanding in complex network/cluster analysis is quite basic, which limits my ability to scrutinize the technical aspects of the study. My comments are therefore directed at the interpretation and scientific merit of the results.

The paper is well written and well organized. The clustering method and results seems to make sense. However, most of the results appear quite trivial (e.g., clusters with ITCZ close to the equator being also more hemispherically symmetric). The added value of the rather complicated novel analysis over more 'traditional' simple methods is either not present or not well communicated. Nevertheless, the authors do refer to the analysis as a first step, establishing the merit of the methodology before examining more broad applications in climate dynamics. In that sense, the consistency of the analysis with known results could be regarded as satisfactory. Some specific comments are provided below.

**Minor comments**

1. It is known that the response of the tropical belt to extratropical SST perturbations lags by 2-4 months. It is not clear to me whether the effect of lagged response is included in the analysis. Since the analysis is based on monthly SST anomalies, it stands to reason that the analysis would be able to convey something about the nature of the lagged response — which at present is not well understood. But this is not discussed in the results.

2. Again, I wonder whether introducing lagged correlations would affect the analysis of tropical vs. extratropical variations. It seems to me that the effects of tropical and extratropical SST anomalies on the tropical rain belt can be thought of as competing paradigms. Tropical SSTs affect the position of the ITCZ via local constraints, whereby the ITCZ resides over the warmest waters. Extratropical SST variations affect the global energy budget, causing the ITCZ to move toward the warming hemisphere. I don't see that the analysis captures this distinction.

3. The 'failure' to diagnose distinctions between the models in response to global warming is somewhat consistent with the minimal zonal-mean ITCZ shifts seen in projections based on comprehensive climate models. The response of the tropical rain belt to global warming is mostly zonally asymmetric, an aspect that was not examined in this work.

**Comments by line number**

36    The energetic framework, as well as SST based arguments have been examined and found to be relevant for time-dependent variations, e.g., during the seasonal cycle (Adam et al. 2016) and in diagnosing potential sources of the double ITCZ bias (Adam et al. 2018). Perhaps this sentence can be clarified or replaced with simply stating that these frameworks are relevant for seasonal or longer climatologies.

201    Typo $CO_2$

---

## Referee Comment (RC2) · Anastasios Tsonis (Referee) · 19 Dec 2020

The authors consider model simulations from 14 GCMs in order to study the dynamics of ITCZ. They use ideas and measures from the theory of complex networks to study the patterns of spatial correlation of monthly surface temperature anomalies as well as the zonal patterns of different topological and spatial network characteristics. The authors of this study are experts in the field, and the study is well executed and well presented. The conclusions appear very solid. I do not have issues with this paper. It should be accepted more or less as is.

---

## Referee Comment (RC3) · Anonymous Referee #3 · 27 Dec 2020

The authors consider functional networks of climate data to understand differences in the dynamics of the ITCZ in state-of-the-art global circulation models. They verify that network modelling correctly identifies that extratropical-tropical connections are important in setting the tropical climate. Also, a strong correlation between Hadley cell strength and ITCZ position is observed in the pattern of connections inside the network. The paper provides important results to understand climate dynamics. My only concern is about the hierarchical clustering, which provides results that depend on the approach considered, such as single or maximum linkage or the Ward method. I suggest the authors to verify whether the results change when these methods are considered in the data clustering step.

---

## Author Comment (AC1) · 3 Feb 2021

**Response to Anonymous Referee # 1**

We gratefully appreciate the generally positive impression of the reviewer regarding the presentation of our study. Furthermore, we thank them for the very interesting and useful suggestions for improving the coherence of the manuscript.

In the following, we present a point-by-point response to the comments and remarks, with the comments of the reviewers shown in blue, italic font.

[Figure]

- *Most of the results appear quite trivial (e.g., clusters with ITCZ close to the equator being also more hemispherically symmetric). The added value of the rather complicated novel analysis over more 'traditional' simple methods is either not present or not well communicated. Nevertheless, the authors do refer to the analysis as a first step, establishing the merit of the methodology before examining more broad applications in climate dynamics. In that sense, the consistency of the analysis with known results could be regarded as satisfactory.*

As also indicated by the reviewer at the end of their comment, our present analysis is meant to be a proof-of-concept that network approaches, which have not yet been applied for this purpose, can provide a tool for better understanding the climate dynamics of tropical rain belts. That is, we show that network approaches can offer a complementary perspective compared to well-established approaches, such as those emphasizing interhemispheric contrasts in SST and energy fluxes, as well as the energy input at the equator. In this sense, the fact that some of the results might appear trivial is at the same time reassuring.

On the other hand, we also agree that the potential for added value should be more clearly articulated wherever possible, even if this potential is not yet realized in our present work. One example is the fact that the networks based on intra-tropical connections only fail to capture model differences in the ITCZ position in the control climate (line 295 of the submitted manuscript). In fact, this is different from the well-established approaches mentioned above, which in the aquaplanet context rely on zonal-mean quantities, and indicates that zonal variations, such as those generated by tropical waves or local SST patches can play an important role even in the aquaplanet setup. Another example is the possibility to expand the methodology so that the networks include other fields or represent lead-lag relationships between tropical and extratropical SST (see below).

We will revise our manuscript to clarify the aim and scope of the present work, and to highlight potentials for future work.

- *It is known that the response of the tropical belt to extratropical SST perturbations lags by 2-4 months. It is not clear to me whether the effect of lagged response is included in the analysis. Since the analysis is based on monthly SST anomalies, it stands to reason that the analysis would be able to convey something about the nature of the lagged response — which at present is not well understood. But this is not discussed in the results.*

  *Again, I wonder whether introducing lagged correlations would affect the analysis of tropical vs. extratropical variations. It seems to me that the effects of tropical and extratropical SST anomalies on the tropical rain belt can be thought of as competing paradigms. Tropical SSTs affect the position of the ITCZ via local constraints, whereby the ITCZ resides over the warmest waters. Extratropical SST variations affect the global energy budget, causing the ITCZ to move toward the warming hemisphere. I don't see that the analysis captures this distinction.*

In the following, we will answer both comments together as they are closely related.

Our network analysis is solely based on instantaneous correlations between tropical and extratropical SSTs. The fact that extratropical SSTs and the ITCZ position are out of phase is an interesting direction for future work that will be pointed out more clearly in the revised discussion and conclusion section. However, studying this aspect seems not trivial, as one would for example need to decide how to blend the phase shift between tropical and extratropical SST anomalies when constructing the correlation matrix for the network analysis.

(As a side note, despite an extensive literature search we were unable to find studies that explicitly show that the ITCZ lags extratropical SST changes by a couple of months. While this is plausible intuitively, we would gratefully appreciate if the reviewer could point us to specific studies on this subject. Our own search only resulted in studies that emphasized the time-mean response or responses beyond 1 year after the perturbation. Also, in observations the ITCZ

leads extratropical SST over the course of the seasonal cycle, as shown, e.g., in Fig. 6 of Chiang and Friedman, Annu. Rev. Earth Planet. Sci. 2012. 40:383–412.)

- *The 'failure' to diagnose distinctions between the models in response to global warming is somewhat consistent with the minimal zonal-mean ITCZ shifts seen in projections based on comprehensive climate models. The response of the tropical rain belt to global warming is mostly zonally asymmetric, an aspect that was not examined in this work.*

  We agree that in comprehensive models with realistic present-day boundary conditions, the zonal variations in the tropical rainfall response to warming can make it difficult to extract a meaningful zonal-mean response. However, this issue should be circumvented in the TRACMIP models as their aquaplanet boundary conditions are zonally symmetric. The 'failure' of our network approach to distinguish model differences in the response must thus have a different origin and indicates that the climate change response of the SST networks is *not* tightly linked to the ITCZ climate change response. Although the reasons for the 'failure' remain unclear to us, one possibility might be that unravelling the climate change response would require a different network representation that involves other atmospheric fields in addition to SST, e.g., changes in the vertical profile and gross moist stability of the tropical atmosphere, which have been shown to be able to play an important role. We will more clearly articulate these points in the revised manuscript.

- *Line 36: The energetic framework, as well as SST based arguments have been examined and found to be relevant for time-dependent variations, e.g., during the seasonal cycle (Adam et al. 2016) and in diagnosing potential sources of the double ITCZ bias (Adam et al. 2018). Perhaps this sentence can be clarified or replaced with simply stating that these frameworks are relevant for seasonal or longer climatologies.*

Thanks, and fully agreed! We will adapt the sentence accordingly so as to make clarify this point and to properly characterize the work of Adam et al. (2016, 2018). We will also revise the manuscript to clarify that our network approach links spatial correlation patterns of the global SST field to the time-mean ITCZ position. This should avoid any confusion regarding the fact that the 'traditional' approaches links the time-mean SST field to the time-mean ITCZ position, where the time-mean can be a seasonal mean or a longer time mean.

- *Typo CO2.*

  We will correct this typo in a revised version of our manuscript.

---

## Author Comment (AC2) · 3 Feb 2021

**Response to Referee # 2 (A.A. Tsonis)**

We gratefully appreciate the very positive impression of Prof. Tsonis regarding the presentation of our study, and thank him for recommending to accept our manuscript for publication in its present form.

---

## Author Comment (AC3) · 3 Feb 2021

**Response to Anonymous Referee # 3**

We gratefully appreciate the generally positive impression of the reviewer regarding the presentation of our study. Furthermore, we thank them for the useful suggestions to further elaborate on the clustering method considered in our study.

Comment of the reviewer: *My only concern is about the hierarchical clustering, which provides results that depend on the approach considered, such as single or maximum linkage or the Ward method. I suggest the authors to verify whether the results change*

*when these methods are considered in the data clustering step.*

We thank the reviewer for giving us the opportunity to comment further on the choice of our clustering method.

In our work, we have utilized hierarchical clustering with a single linkage approach for grouping different models in a statistically meaningful way. This choice has been motivated by the fact that we have been interested in merging the most similar models into the same groups. With only a low number of models included in the study, employing the single linkage method ensures that models exhibiting comparable zonal network distributions end up in the same model cluster.

In addition to the single linkage approach, we have also employed the widely used alternatives of complete linkage and average linkage, i.e., grouping according to the largest or mean distances among pairs of elements from the clusters to be merged instead of the minimum distance used for the single linkage approach. In our original manuscript, we had decided to not report the corresponding results, since those different options have led to rather dissimilar dendrograms and, hence, group structures. The finding is however not surprising at all, but a common feature of hierarchical cluster analysis. In the following, we will further detail the observed differences.

In general, unlike the single linkage method, the two other alternatives resulted in clusters that have been hardly interpretable to us. As the most striking feature, they essentially separated individual outliers from a large group including the remaining models. In this regard, at least a considerable part of the obtained clusters have been found robust under those variations of the methodology, i.e., some combinations of models always appeared in the same group independent of the linkage strategy used.

For complete linkage clustering (which uses the farthest pairs of models as a criterion for grouping), the outliers essentially determine the clustering procedere and prevent certain cluster configurations to become resolved by the method. For the average linkage approach, it is likely that the specific distribution of similarity scores (see Fig. 5) causes the dissimilar outcome as compared to the single linkage clustering. Specifically, models belonging to the same cluster under the single linkage approach only show minor differences in their mutual similarity. By considering the average of all pairwise similarities, models with rather low mutual similarity potentially have a large influence on the resulting cluster structure.

Besides those different hierarchical clustering approaches, there would be further methodological alternatives to hierarchical (like Ward's method mentioned by the reviewer, or the centroid method) as well as non-hierarchical clustering methods (including partitioning approaches like k-means, spectral clustering, and many more options). Given the relatively small size of our set of models, it appears to us being of less interest to test a larger variety of possible approaches and select one that provides the "optimal" group configuration according to some established statistical model selection criteria.

We will use the opportunity of revising our manuscript to include a more thorough discussion of this aspect as a part of Section 2.4 focusing on the robustness of our findings.

---

## Author Response (AR2)

Frederik Wolf
Potsdam Institute for Climate Impact Research
PO Box 601203, 14412 Potsdam, Germany
email: frederik.wolf@pik-potsdam.de

[Figure]

Potsdam, February 21, 2021

Dear Dr. Messori,

we have gratefully taken the opportunity to discuss the mentioned literature and prepare supplementary information containing the details and figures of our robustness tests.

Regarding the literature, we want to thank you for pointing us to the papers. We have carefully checked them and have included Woelfe et al. 2015 and Adam et al., 2018 in our conclusion section. The paper of Kang and Xie (2014) only considers the time mean SST abd ITCZ and so we did not include it. We do believe the reviewer instead meant Kang et al., 2014, Clim Dyn, 42:20332043, doi:10.1007/s00382-013-1863-0. Although this paper very briefly touches upon the transient response to extratropical heating changes, the time lead of extratropical SST is not as clear as in Woelfe et al. We therefore did not include Kang et al. 2014.

To complement our analysis, we have now included the figures illustrating the results of the robustness tests. In that scope, we have also identified a minor error in the complete linkage scheme analysis, which actually led to an increased similarity between the employed single linkage approach. Therefore, we have conducted slight changes in that direction in the main manuscript aswell (page 11, lines 263-272).

On behalf of the authors,

Sincerely,

Frederik Wolf